# Mycoplasma genitalium adhesin P110 binds sialic-acid human receptors

David Aparicio[1], Sergi Torres-Puig [2], Mercè Ratera [1], Enrique Querol[2], Jaume Piñol[2], Oscar Q. Pich [2] & Ignacio Fita[1]

Adhesion of pathogenic bacteria to target cells is a prerequisite for colonization and further infection. The main adhesins of the emerging sexually transmitted pathogen Mycoplasma genitalium, P140 and P110, interact to form a Nap complex anchored to the cell membrane. Herein, we present the crystal structures of the extracellular region of the virulence factor P110 (916 residues) unliganded and in complex with sialic acid oligosaccharides. P110 interacts only with the neuraminic acid moiety of the oligosaccharides and experiments with human cells demonstrate that these interactions are essential for mycoplasma cytadherence. Additionally, structural information provides a deep insight of the P110 antigenic regions undergoing programmed variation to evade the host immune response. These results enlighten the interplay of M. genitalium with human target cells, offering new strategies to control mycoplasma infections.

[1] Instituto de Biología Molecular de Barcelona (IBMB-CSIC) and Maria de Maeztu Unit of Excellence, Parc Científic de Barcelona, Baldiri Reixac 10, 08028 Barcelona, Spain. [2] Institut de Biotecnologia i Biomedicina and Departament de Bioquímica i Biologia Molecular, Universitat Autònoma de Barcelona, 08193 Bellaterra, Barcelona, Spain. Correspondence and requests for materials should be addressed to O.Q.P. (email: oscar.quijada@uab.cat) or to I.F. (email: ifrcri@ibmb.csic.es)

Bacterial pathogens rely on dedicated surface structures capable of mediating specific or nonspecific adhesion to host target receptors to establish successful infections[1,2]. Sialic acids are negatively charged hydrophilic carbohydrates that occupy external positions in mucosal surfaces and secreted sialoglycoproteins from eukaryotic cells. Given their accessibility and ubiquitous distribution, a variety of microbial strategies target host sialic acids for adherence, mimicry, and degradation[3]. Regarding adhesion, it is known that the SabA adhesin of *Helicobacter pylori*[4] or the phylogenetically related streptococcal adhesins GspB, Hsa, and SrpA[5] bind to sialylated ligands. Similarly, several Mycoplasma species have been shown to interact with sialylated glycoproteins[6–8] and it is well-established that the respiratory pathogen *Mycoplasma pneumoniae* attaches to human erythrocytes through sialylated oligosaccharides of the Ii antigen type[9,10].

*Mycoplasma genitalium* is a facultative intracellular pathogen closely related to *M. pneumoniae*[11,12]. It has been implicated in urogenital pathologies such as urethritis in men and cervicitis and pelvic inflammatory disease in women[13]. Upon interaction with host cell receptors, some *M. genitalium* cells invade the host cells by an unknown mechanism[12,14]. Intracellular location of *M. genitalium* constitutes an important strategy to avoid the host immune system and likely contributes to the establishment of persistent infections[15]. In addition, the rapid emergence of antibiotic resistance in *M. genitalium*[16–18], emphasizes the urgency for the development of new therapeutic strategies. In this sense, a better understanding of the cytadherence mechanics could facilitate the implementation of more effective antimicrobial approaches based on antiadhesion therapies[19].

The major cytadhesins of *M. genitalium*, P140 (1444 residues) and P110 (1052 residues)[20] show an extensive immunological cross-reactivity with the main cytadhesins of *M. pneumoniae*, namely P1 and the P90/P40 pair, respectively[21]. P140 and P110, which have been shown to be reciprocally stabilized, interact to form a transmembrane complex, called "Nap", that accumulates at a polar structure known as the terminal organelle[22]. The term Nap was originally coined by Dr. Tully and colleagues to designate the short fibers decorating the outer surface of the terminal organelle of *M. genitalium*[23]. The Nap is expected to be instrumental for adherence and gliding motility in mycoplasmas

belonging to the *pneumoniae* cluster[24,25]. In addition to their role in adherence and locomotion, P140 and P110 have been also implicated in the duplication of the terminal organelle during cell division[26], and they constitute the main target of host antibodies during infection[21,27–29]. For this reason, *M. genitalium* has evolved a refined strategy to generate antigenic variants of these two immunodominant proteins. Scattered throughout the *M. genitalium* genome there are nine repeat regions, designated as MgPar, that contain sequences with homology to the cytadhesin genes[30]. Recombination between the cytadhesin genes and homologous MgPar sequences, provides a virtually unlimited collection of antigenic variants[31,32].

In the present work, we have determined the crystal structure of the P110 adhesin alone and in complex with sialylated oligosaccharides with surface plasmon resonance (SPR) providing the respective equilibrium dissociation constants ($K_D$). The structural information guided the design of P110 variants that present important cytadherence defects in vivo. Overall, our findings substantiate a central role for P110 in *M. genitalium* cytadherence and identify the binding domain of this adhesin to sialylated receptors. Moreover, the topography of P110 provides important clues to the understanding of the antigenic properties of this immunodominant protein.

## Results

**Crystal structure of P110 from *M. genitalium*.** Crystals were obtained from the extracellular region of *M. genitalium* adhesin P110 (erP110) (residues 23–938). Therefore, erP110 does not include either the residues corresponding to the signal secretion peptide (residues 1–22) or the transmembrane and intracellular regions (938–1052) at the N- and C-ends of P110, respectively (Supplementary Figure 1). The erP110 structure was solved by single-wavelength anomalous diffraction (SAD) and density modification at 2.95 Å resolution (see Experimental procedures) (Fig. 1a). Six selenium sites were located in the crystal asymmetric unit, which contains only one erP110 subunit with solvent occupying 67% of the volume. Initial maps allowed building a partial model that was completed and refined, with data at 2.7 Å, giving agreement factors $R$ and $R_{free}$ of 18.4/22.9, respectively (Supplementary Table 1).

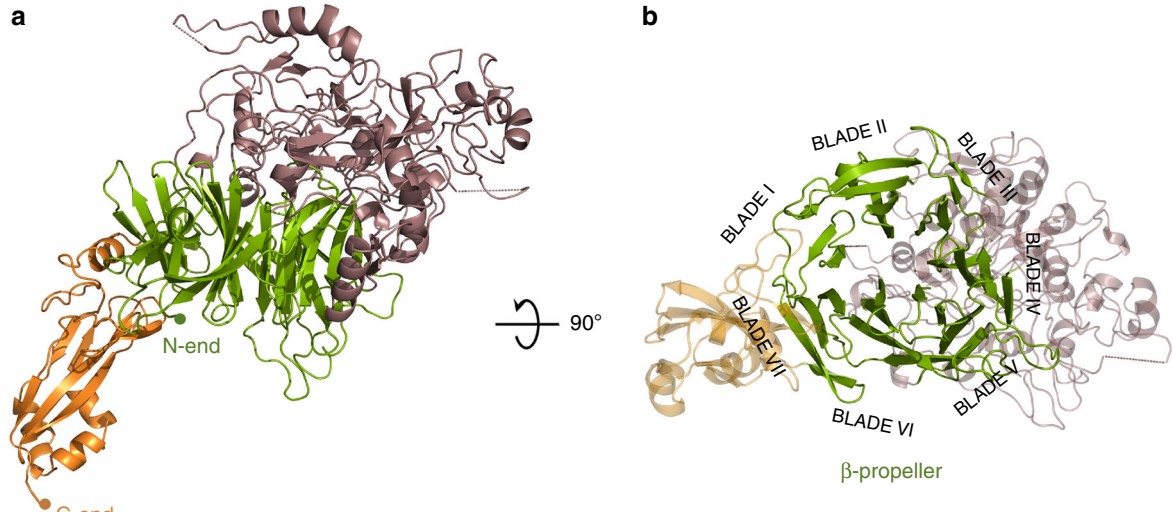

**Fig. 1** Overall structure of P110. **a** Two views, 90° apart from each other, of the extracellular region of P110 that is formed by a large N-domain, with a seven blade β-propeller (green), the crown (brown), and the C-domain (orange). In the right side panel the view is along the central axis of the β-propeller. The situation of the seven blades in the propeller is explicitly indicated showing that the two terminal blades I and VII are close to the C-terminal domain and opposite to the crown

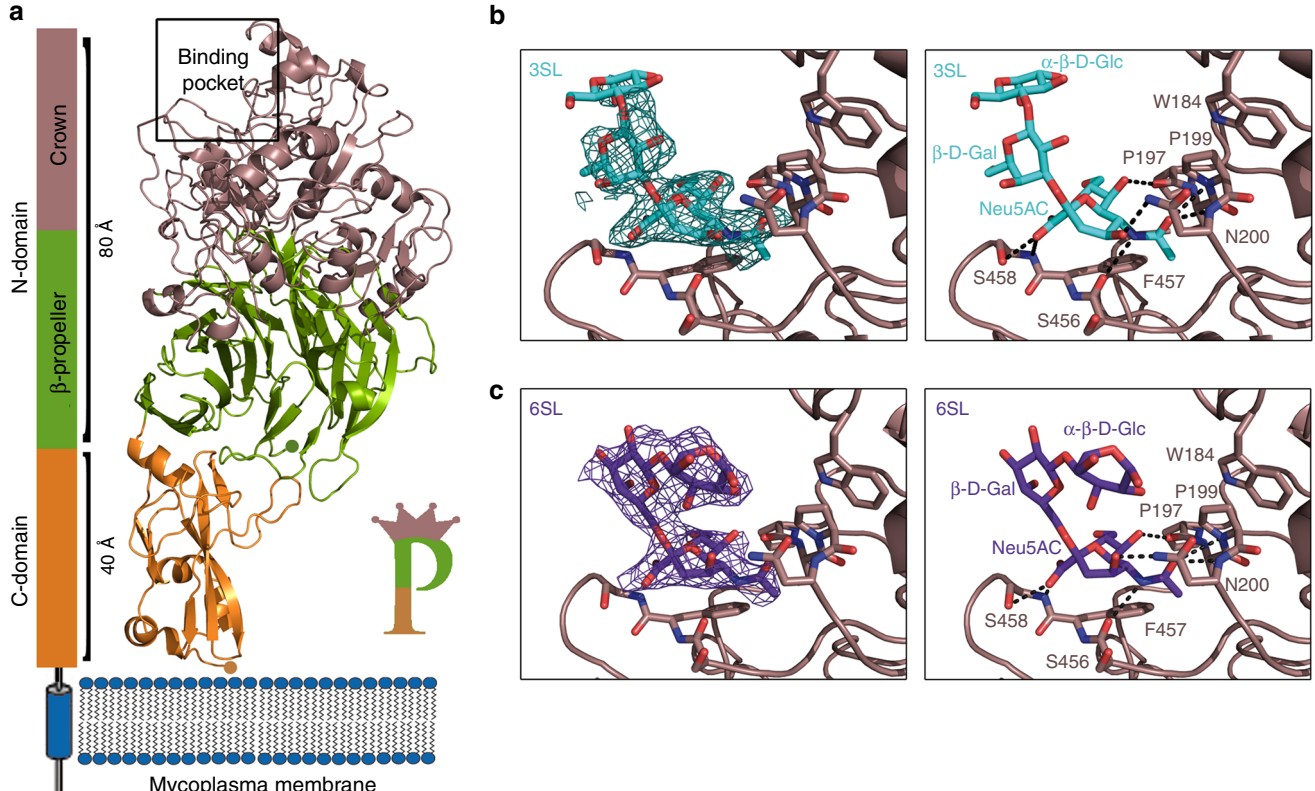

**Fig. 2** Binding of sialic acid oligosaccharides to P110. **a** Schematic representation of the disposition of P110 with respect to the mycoplasma membrane (same color code as in Fig. 1). The predicted transmembrane helix follows in sequence to the C-domain, which accordingly is expected to be close to the cell membrane. The overall structure of P110 can be sketched as a capital letter **P** with the sialic binding site (indicated in the figure with a rectangle) located in the crown, away from the cell membrane. Detail of the binding to P110 of sialic acid oligosaccharides **b** 3SL and **c** 6SL. Left panels show the electron density corresponding to oligosaccharides in a sigma weighted (Fo–Fc) omit map at two sigma. The binding site and the binding interactions with P110 for both oligosaccharides are very similar, but not identical

The structure of erP110 consists of a large N-domain (residues Ala25–Asp812) and a smaller C-domain (residues Thr813–Leu936) (Fig. 1a). The N-domain presents a β-propeller topology with seven consecutive blades corresponding to β-sheets I–VII (Fig. 1a). The propeller organization results in every β-sheet being in contact with the β-sheet next in sequence (I with II, II with III, and so on until closing the propeller, VII with I). The β-sheets each have four antiparallel strands, except β-sheet I that has three. Connections between β-strands vary widely in length and accordingly the number of residues that each β-sheet spans is very diverse, ranging from 26 to 197 (Thr27–Asp53 (I), Ala63–Leu132 (II), Ser300–Trp341 (III), Ala382–Pro578 (IV), Asn624–Phe686 (V), Thr697–Thr772 (VI), and Gln774–Asp812 (VII)) (Supplementary Figure 2). The length of links between β-sheets is also very diverse, going from zero residues, between sheets VI and VII, to 168 residues, between sheets II and III. Residues from the long connections, both between β-strands and β-sheets, cluster together creating a crown-like structure above β-sheets III–V. The centre of the propeller is not occupied by protein residues resulting in a cavity connecting the two faces of the propeller, that runs parallel to the seven surrounding β-strands, one from each of the seven β-sheets. Search for the closest structural homology with DALI server[33] resulted in a root mean square deviation (r.m.s.d.) of 3.6 Å for 255 residues with Virginiamicyn B Lyase (Supplementary Figure 3), an antibiotic degradation bacterial enzyme belonging to the seven bladed β-propeller superfamily. The C-domain of erP110 presents a compact fold, with a β-barrel of five antiparallel strands, a β-hairpin and four small α-helices. For this C-domain only weak

structural homologies were found by DALI, with Z scores below 4 and less than 50% equivalent residues. The C-domain protrudes radially (~40 Å) from the propeller interacting with β-sheets I, VII and also, though less extensively, with β-sheet II.

The overall shape of erP110 can be described as a capital letter **P** (of ~120 Å in length), with the round and leg parts of the **P** corresponding to the N- and C-domains, respectively (Fig. 2a). In this representation, the C-end of the C-domain is located at the very bottom of the **P**, while the crown is at the top of the **P** above the β-sheets of the propeller that are located opposite to the C-domain. The C-end of erP110, predicted to be followed by the transmembrane helix (Supplementary Figure 1), is expected to be close to the outer surface of the mycoplasma membrane (Fig. 2a). Electron microscopy reconstructed volumes of Naps, obtained at 15–20 Å resolution by negative staining of purified Naps and also by cryo-tomography of mycoplasma cells and subtomogram averaging of Naps present a large globular region that connects with the cell membrane by a stalk[34]. These reconstructed volumes of the Nap, with a size that corresponds to about the number of residues in the extracellular regions of two P110-P140 hetero-dimers, present a shape that can match well with the organization of erP110 with the bulky, β-propeller, domain, and the small and elongated C-domain.

Electron density is well defined for all the C-domain residues. On the contrary, density is poor in four regions of the N-domain where ~25 residues could not be modeled. These 4 disordered regions correspond to solvent exposed flexible loops in the crown (around Gly257, 413–416, 468–477, and 593–602), including a stretch of 11 contiguous serine residues (409–419). The

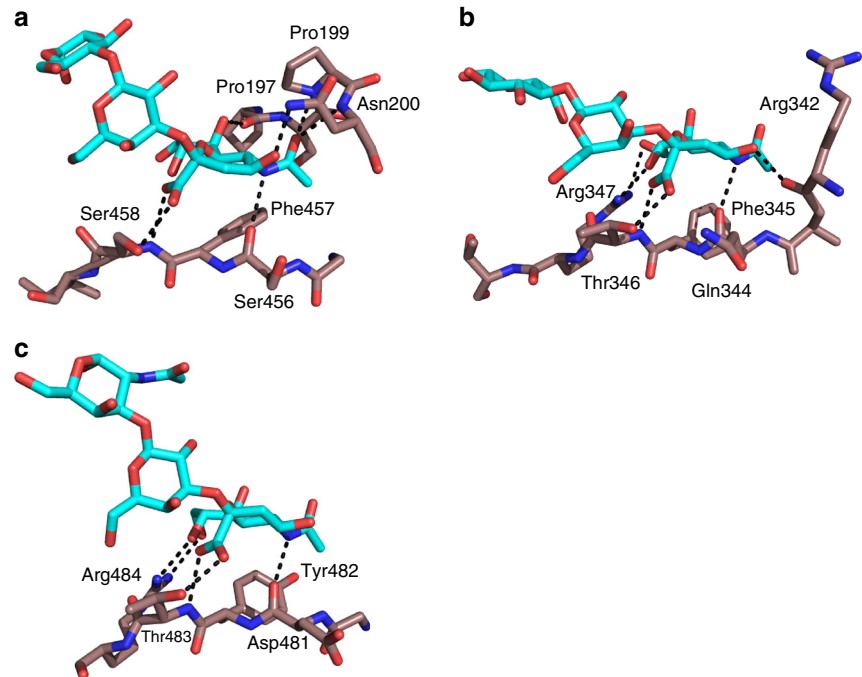

**Fig. 3** Comparison of sialic binding sites in bacterial adhesins. Binding of the oligosaccharides neuraminic acid moiety to **a** adhesin P110 from *M. genitalium* and to the serine-rich repeat (SRR) adhesins from **b** SrpA[67] (PDB accession code 5IJ1), and **c** GspB[68] (PDB accession code 5IUC). The extended tripeptide X-Tyr/Phe-Ser/Thr (residues 456–458 in P110) appears to be a common binding motif for sialic acid. Carbon atoms for the oligosaccharides are represented in light blue, while for amino acid residues are represented in brown. Nitrogen and oxygen atoms are depicted in dark blue and red, respectively

disordered region around residue Gly257 corresponds to a large insertion in P90/P40 (MPN142), the orthologous protein of P110 in *M. pneumoniae* (Supplementary Figure 2). It is interesting to note that three of the four disordered regions, with the exception of region (593–602), correspond to MgPar repeats of the P110 gene. In fact, the poly-serine (poly-"AGT") tract is located within the large repeat referred as KLM (residues 42–517), with homology to the MgPars 2, 8, and 9, which contain 16, 10, and 9 "AGT" triplets, respectively[31]. Crystal packing leaves the crown of the N-terminal domain mostly free of interactions with other subunits. Accessibility of the crown has been instrumental in obtaining crystals of complexes of erP110 with sialylated oligosaccharides (see below).

**Complexes of P110 with sialylated oligosaccharides 3SL and 6SL.** As mentioned above, sialylated oligosaccharides have been shown to be the main host receptors in *M. pneumoniae* and in some other motile mycoplasmas[35,36]. Therefore, we focused on sialylated oligosaccharides when starting to look for possible interactions of erP110 with cell receptors of *M. genitalium*. Co-crystallization of erP110 with either 3′-sialyllactose (3SL: where the neuraminic acid forms an α2–3 linkage to a lactose monosaccharide) or 6′-sialyllactose (6SL: with an α2–6 linkage) resulted in complexes of both oligosaccharides with erP110, as determined at 2.2 and 2.5 Å resolution, respectively (Experimental procedures section). The oligosaccharides binding site, the same for both complexes, is located in an indent of erP110 in the upper part of the crown (Fig. 2a). In both complexes only the neuraminic acid moiety presents direct interactions with two short stretches of the protein: Pro197–Ser198–Pro199–Asn200 and Ser456–Phe457–Ser458. The carboxylate group from the neuraminic acid forms two hydrogen bonds with Ser458, one with the backbone amino group and the second with the side chain oxygen. The neuraminic acid also makes another hydrogen bond with the side chain from

Asn200 and four more (three for 3SL) with the main chain. Finally, the methyl group of the neuraminic acid is placed into a hydrophobic pocket close to the Phe457 side chain. Trp184 appears important for shaping the binding pocket, though without interacting directly with the bound oligosaccharides (Fig. 2b, c). In the refined structures, oligosaccharides present different occupancies (~100% for 6SL and ~60–70% for 3SL), which suggest different binding affinities in spite of the fact that neuraminic acid is making about the same interactions in both complexes. The lactose moiety of 6SL is also well defined in the 6SL complex, while density is progressively weaker for 3SL suggesting that, beyond the neuraminic acid, the conformation of this oligosaccharide can fluctuate. The third monosaccharide approaches the protein in 6SL, while it is projected towards the solvent in 3SL (Fig. 2b, c).

No structural similarities were found between P110, with a serine content of 12%, and the serine-rich repeat (SRR) family of adhesins[33]. However, we observed a striking correspondence between the interactions and the conformation of the sialic acid moiety bound to P110, and the ones seen in the SRR proteins GspB (*Streptococcus gordoni*) and SrpA (*Streptococcus sanguinis*) involving mainly a tripeptide X-Tyr/Phe–Ser/Thr (residues 456–458 in P110) (Fig. 3). The carboxylic group of sialic acid points toward the third residue in the tripeptide (Ser/Thr), while the amide group of sialic hydrogen bonds to the carbonyl group of the first residue. The Ser/Thr position has been shown to be critical for binding with single point mutations in SrpA presenting a dramatic decrease in binding to human platelets[37]. For P110, the *M. genitalium* strain variant with the S458D substitution was unable to produce protein (see below). Moreover, in the three structures the sialic methyl group is placed into a hydrophobic pocket created by the aromatic residue of the tripeptide (Fig. 3). Residues outside the tripeptide can contribute to the binding of sialic oligosaccharides through specific interactions that differ among different proteins (Fig. 3). Thus,

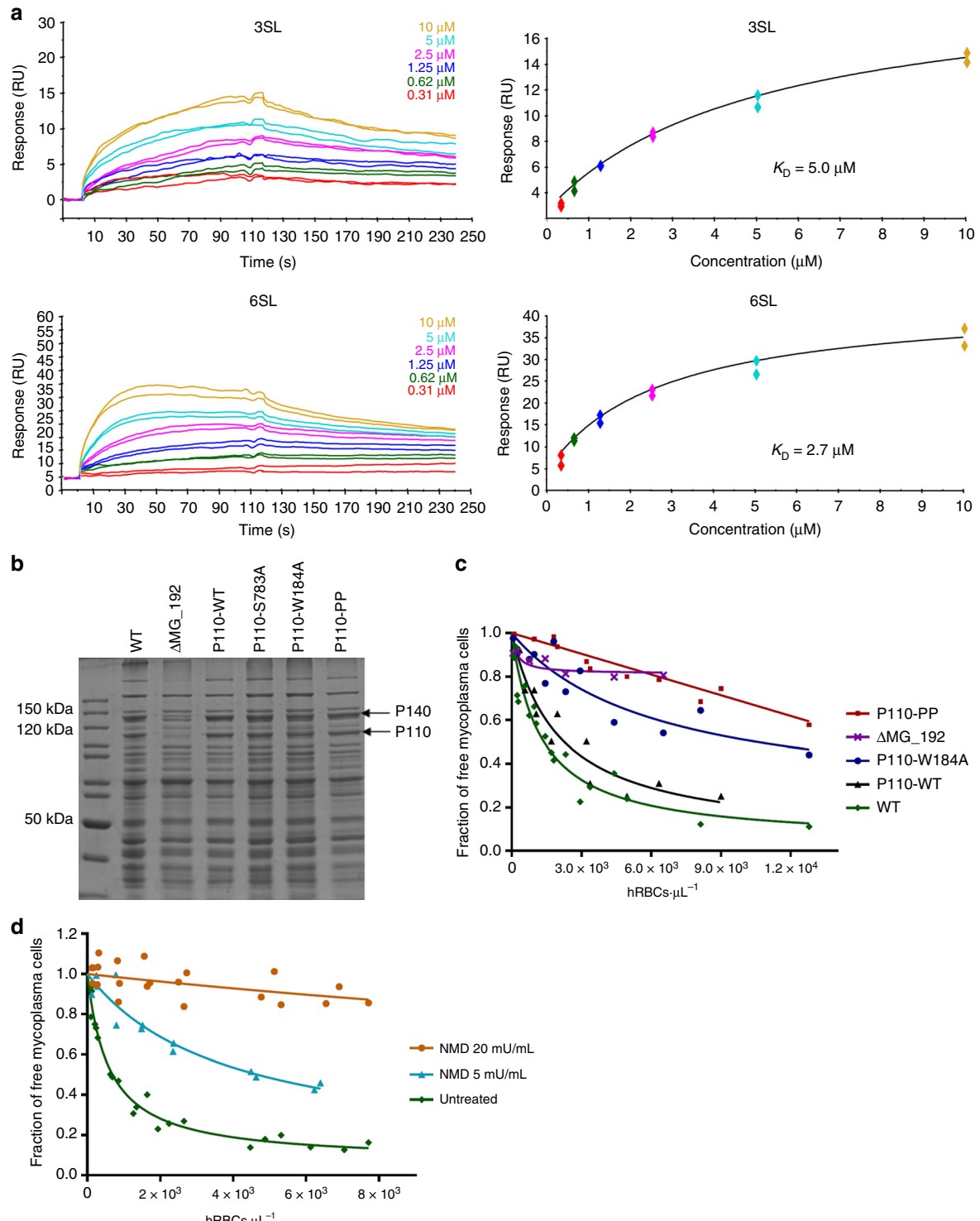

**Fig. 4** Affinity of oligosaccharides by SPR. Characterization of *M. genitalium* P110 variants. **a** SPR sensorgrams showing the binding of the erP110 β-propeller N-domain to immobilized 3SL and 6SL (upper and lower left panels, respectively). Data fitting of duplicates assuming a Langmuir 1:1 affinity model depicting the $K_D$ calculated in steady state for 3SL and 6SL (upper and lower right panels, respectively). **b** SDS–PAGE of whole cell lysates from WT and different P110 mutant strains. **c** HA assays determined by FACS analysis. Graphic depicts inverse Langmuir plots obtained with a fixed amount of mycoplasma cells and increasing amounts of hRBCs. Plots were generated using data from, at least two biological repeats for each strain. **d** HA assays of the WT strain using neuraminidase-treated hRBCs

while a few arginine residues are determinant in SrpA (Arg342 and Arg347) and GspB (Arg484), some prolines (Pro197 and Pro199) seem to be critical in P110 (see below). Affinity between erP110 and the sialic acid compounds was experimentally characterized by Surface Plasmon Resonance using biotinylated 3SL and 6SL immobilized on a streptavidin chip (Fig. 4a). Equilibrium dissociation constants ($K_D$), determined from the steady state binding isotherms using a 1:1 Langmuir model, show

an affinity for 6SL ($K_D = 2.7\,\mu M$) that is almost twice the one found for 3SL ($K_D = 5.0\,\mu M$), which could explain the lower occupancy observed for 3SL in the erP110 complexes.

**Cytadherence of *M. genitalium* strains carrying target mutations in P110.** The structure of P110 in complex with sialylated oligosaccharides revealed the candidate residues for binding. To assess the implication of these residues in cytadherence, we performed target mutagenesis of the MG_192 gene, which codes for the P110 protein, to generate the following variants: W184A, S458D, S783A, P197A-P199A (PP mutant), and deletion of the loop S458-T462 (Δloop mutant). The resulting MG_192 mutant alleles were introduced into *M. genitalium* by transposon delivery (TnPacP110) and selected for by means of the puromycin resistant marker (*pac*). To avoid the presence of two different copies of the MG_192 gene in the chromosome, the different P110 variants were expressed on a ΔMG_192 mutant background[22]. For control purposes, a wild-type copy of the MG_192 gene was also delivered to the strain lacking P110 using the same minitransposon.

Whole cell lysates of the wild-type strain, the ΔMG_192 mutant and representative clones carrying the wild type or mutant MG_192 alleles were analyzed by sodium dodecyl sulfate polyacrylamide gel electrophoresis (SDS–PAGE) (Fig. 4b). We found, as reported[22], that the band corresponding to the P110 protein was missing and the amount of P140 protein was considerably reduced in the protein profile of the ΔMG_192 mutant. In contrast, P110 and P140 levels were restored upon delivery of the MG_192 allele in the P110-WT, P110-S783A, P110-W184A, and P110-PP mutants. However, we did not detect the presence of P110 in the transformants carrying the variant with the S458D substitution or the deletion of residues Ser458 to Thr462 (Δloop mutant). At the light of this finding, the two latter mutants were not further characterized. Surprisingly, while the S458D mutant was produced in vitro similar to the wild-type erP110, the Δloop variant resulted, by gel filtration, in a heterogeneous protein distribution that could not be purified. These observations suggest that the loop is structurally critical, with Ser458 being important for the proper processing and stability of P110 in vivo.

Mycoplasma cytadherence is usually evaluated by their capacity to adhere to red blood cells, which is known as hemadsorption (HA). Therefore, we used flow cytometry to quantitatively determine the HA capacity of the wild-type strain (G37), the ΔMG_192 mutant, and representative clones carrying wild-type or mutant MG_192 alleles (Fig. 4c). Fixed amounts of mycoplasma cells were mixed with increasing concentrations of human red blood cells (hRBCs) and stained using SYBR green[38]. Binding of mycoplasma cells to hRBCs can be modeled in an inverse Langmuir isothermal kinetic function. Preincubation of hRBCs with neuraminidase, an exosialidase which cleaves α-ketosidic linkage between the sialic acid and an adjacent sugar residue, abrogated or reduced considerably the HA capacity of *M. genitalium* (Fig. 4d). This result demonstrates that binding of this human pathogen to hRBC is mediated through the interaction with sialylated receptors. As expected, cells from the reconstituted P110-WT strain displayed a HA capacity similar to that of WT cells. Likewise, HA capacity of the P110-S783A mutant was comparable to that of the P110-WT strain (data not shown). In contrast, the HA capacity of the P110-W184A mutant was considerably reduced, although some weak binding was still observed. Moreover, our results indicate that cells of the P110-PP strain barely bind hRCBs and accordingly, HA data did not fit in a Langmuir isothermal curve. This behavior is expected for mutants with severe cytadherence defects. Therefore, as indicated by the structural data, our in vivo analysis corroborates that Trp184, Pro197, and Pro199 play a key role in the interaction of *M. genitalium* with host cell receptors.

**Binding of potassium to P110.** The structures of complexes with oligosaccharides also showed a potassium ion bound to the C-domain, which was absent in the apo-erP110 structure (Fig. 5). Whereas potassium was not used in the crystallization of apo-erP110, potassium was added as tri-potassium citrate during co-crystallization with both oligosaccharides and its presence in the crystal confirmed by X-ray fluorescence spectroscopy (Supplementary Figure 4). The potassium ion binding site is located in a hydrophilic cage of the C-domain close to the interface with the N-domain (Fig. 5a). Potassium is coordinated with three oxygens of the main chain (from residues Thr831, Arg834, and Gly839), with two water molecules and with one of the carboxylate oxygens from Asp836 (Fig. 5b). The cage is covered by the Tyr830-side chain, which maintains all the carbon atoms of the phenolic ring at ~4 Å from the potassium. In the absence of potassium, the binding cage opens by reorienting the Tyr830 side chain, and the Asp836 side chain rotates outside the cage pointing its carboxylic group toward the displaced side chain of Arg834 from a crystal twofold related erP110 subunit (Fig. 5c). These changes also result in some rearrangements of neighbour residues, in particular Pro877 and Pro926. The relative orientations of the N-domain and the C-domain are essentially identical (with differences smaller than 2°) in the four erP110 structures determined. Though many roles have been described for potassium ions, its function in P110 activity is unclear[39,40].

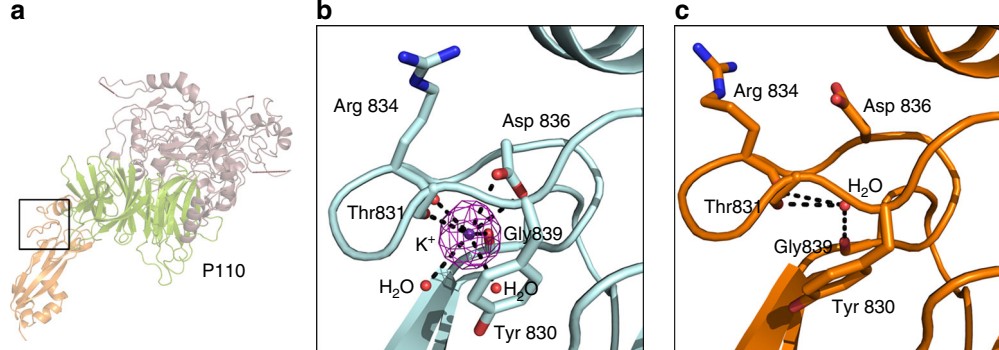

**Fig. 5** Binding of Potassium to P110. **a** Overall views of the potassium binding site (indicated by a rectangle) in the C-terminal domain of erP110. Detail of the potassium binding site in the presence (**b**) and in the absence (**c**) of potassium. Electron density corresponding to the potassium is shown with a sigma weighted (Fo–Fc) omit map at four sigma

## Discussion

Mycoplasmas are arguably among the most successful pathogens, establishing intimate interactions with epithelial cells and causing chronic diseases in humans and a wide range of animals. However, so far, the mechanisms employed by these microorganisms to colonize their hosts and subvert the immune system are still poorly understood. It is noteworthy that several mycoplasma species invade its target cells and survive within the intracellular environment, which provides a safe harbor from host defenses[15]. *M. genitalium* is a motile bacterium that attaches to human epithelial cells of the genital tract by means of a specialized tip structure that agglutinates its main cytadhesins and a bunch of cytoskeletal proteins[22,41]. *M. genitalium* cytadhesins are multipurpose proteins and, besides its key role in cell adherence, they are important for locomotion and cell invasion. Therefore, they represent bona fide virulence factors crucial for colonization, dissemination, and persistence. To gain understanding into the function of the cytadherence machinery of mycoplasmas, in this study we have determined the crystal structure of P110, a major adhesin of *M. genitalium* and an immunodominant protein of this emergent sexually transmitted pathogen.

The structure of the extracellular region of P110 is composed by two domains, the N- and C-domains, very different in size, with an overall shape and domain organization that can be sketched as the body and the stalk of capital letter **P** (Fig. 2a). The stalk corresponds to the small C-domain that is followed in sequence by a predicted transmembrane helix, which imposes the proximity of this C-domain to the cell membrane. The C-domain adopts a previously unreported compact fold and presents a potassium binding site close to interface with the β-propeller N-domain. Binding of potassium results in structural changes in a few residues, but possible functional of potassium in P110 remain unclear. The large N-domain, with more than 800 residues, presents a topology consistent with a seven blade β-propeller that is closely related to the one found in a large variety of protein families including lyases, oxidoreductases, and hydrolases[42]. The highest similarity, according to the DALI server, is with the antibiotic degradation enzyme Viginiamycin B lyase, though none of the catalytically characteristic residues is found in P110. Interestingly, the β-propeller topology is also found in sialic acid binding proteins such as neuraminidases[43]. Within the N-domain of P110, long connections between some blades and also between some β-strands create a structure, referred as "the crown", containing a binding site for sialic acid oligosaccharides, as conclusively shown by the structures of the complexes of P110 with both 3′- and 6′-sialyllactose oligosaccharides. Therefore, in contrast to what is found in neuraminidases[43], the sialic acid binding pocket identified in P110 is located outside the β-propeller. In addition, we have not detected sialidase activity in vitro using the recombinant P110 protein (data not shown). Altogether, this information suggests that the β-propeller found in P110 has mainly a structural role as it has been observed in other proteins such as integrins[44,45]. Binding of P110 to sialylated oligosaccharides is characterized by interactions of the neuraminic acid moiety with the tripeptide Ser456–Phe457–Ser458. These interactions are shared with other bacterial adhesins binding to sialylated oligosaccharides by the tripeptide binding motif X-Phe/Tyr–Ser/Thr. Besides this common pattern of interactions, binding to the neuraminic acid moiety is reinforced in P110 with specific interactions involving mainly residues Pro197–Ser198–Pro199–Asn200. The sialylated oligosaccharide binding site in P110 was confirmed, by mutagenesis and HA assays, to be crucial to initiate attachment to human cells. Therefore, sialylated oligosaccharides appear to be the main cell host receptors in *M. genitalium*

infection. Results presented in this study also provide the first experimental evidence for a protein, P110, binding to sialylated compounds in *M. genitalium*. Remarkably, sequence alignments indicate that important residues at the binding site in P110 are conserved in its homologous protein P90/P40 (MPN142) of *M. pneumoniae* (Fig. 6a). So far, it is unclear whether P140, the partner of P110 in the Nap complex, can also bind sialylated compounds, but experiments performed with P1 (MPN141), homologous to P140 in *M. pneumoniae*, failed to detect interactions of this protein with sialylated glycoproteins[25].

In the structure of P110, the MgPar variable regions KL (residues 42–294) and LM (462–517) correspond to stretches of the N-domain highly accessible to the solvent (Fig. 6b). Remarkably, within these variable regions, residues important for oligosaccharide binding appear to remain mostly constant as expected if the functionality of the binding site has to be preserved. The poly-"AGT" tract of variable length, found within the extended KLM repeat, also protrudes as a disordered loop from the P110 surface in the vicinity of the oligosaccharide binding site. Therefore, all the potential natural variants of P110 generated by recombination result in a highly variable solvent exposed surface surrounding an exquisitely conserved receptor binding pocket (Fig. 6a, b). The structural design would protect this essential functional site of P110 by diverting the immune system response to the variable regions, a strategy reminiscent to what had been proposed for pathogenic viruses[46,47]. The structural context provides a rationale for the striking capability of *M. genitalium* to generate extensive variations of this immunodominant protein[29,48–50]. Together with previous studies of adhesin based vaccines[51,52], these results offer alternative strategies to block the first stages of the infection by *M. genitalium*, which is evolving into a so-called superbug because of its antibiotic resistance and considered a global public health threat[53].

## Methods

**Cloning, expression, and purification of P110**. Region corresponding to the MG_192 gene from *M. genitalium* (strain G37, residues 23–938) was amplified from synthetic clone (Genscript) using primers P110F and P110R as Forward and Reverse, respectively (Supplementary Table 2). The PCR fragments were cloned into the expression vector pOPINE[54] (gift from Ray Owens Addgene plasmid #26043) to generate a C-terminal Histidine tagged protein. The recombinant protein was obtained after expression in B834(DE3) cells (Merck) at 20 °C o/n upon induction with 1 mM IPTG at 0.6 $OD_{600}$. Cells were harvested and lysated in 1× PBS buffer by sonication. Subsequently, cell extract was centrifuged at 20 krpm at 4 °C and the supernatant applied to a 5 mL Histrap (GE Healthcare) column equilibrated with 1× PBS as a binding buffer and 1× PBS, 500 mM imidazole as an elution buffer. Soluble aliquots of $His_6$-tagged P110 were pooled and loaded onto a HiLoad Superdex 200 16/60 column (GE Healthcare) in buffer consisting of 50 mM Tris pH 7.4 and 150 mM NaCl. SeMet-labeled P110 was produced with SeMet$^{TM}$ media (Molecular dimensions$^{TM}$) following the protocol provided in the MD12-500 Kit. Purification of SeMet-labeled protein was conducted using the same protocol described for native P110 protein.

**Crystallization and preparation of complexes with oligosaccharides 3SL and 6SL**. Screening for initial crystallisation conditions were performed with 150 nL droplets of 96-well plates on Cartesian robot. Crystals from native and SeMet-labeled P110 recombinant protein were prepared mixing 1 + 1 µL of P110 protein (13.2 mg/ml) and reservoir solution containing 11% PEG 8000, 0.1 M Imidazole pH8, 0.2 M Calcium Acetate at 20 °C temperature. In order to obtain both 3SL and 6SL complex, native P110 was incubated with 10 mM of each oligosaccharide 30 min at 20 °C before mixed with reservoir solution containing 21% PEG 3350, 0.2 M tri-potassium citrate. All Crystals were flash-frozen in liquid nitrogen with 20% glycerol as a cryoprotectant.

**X-ray data collection and structure determination**. X-ray diffraction experiments were performed at Xaloc Beamline (ALBA, Spain) for native, SeMet labeled and 3SL crystals. 6SL crystal diffraction experiments were measured at ID23-1 Beamline (ESRF, Grenoble). SAD data from SeMet crystal was collected at Selenium-K edge energy. Data were processed with Xia2[55] using XDS[56], Aimless

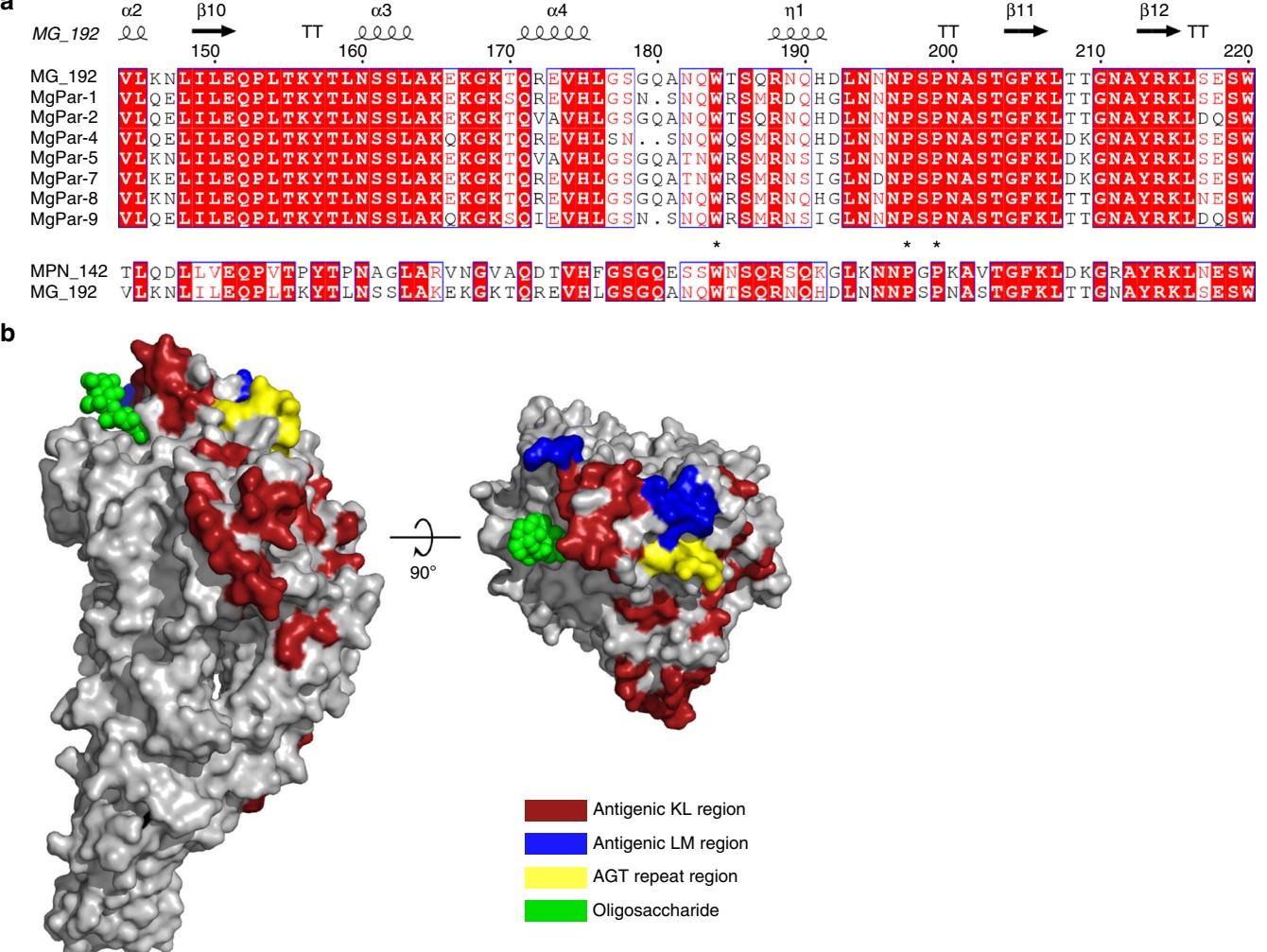

**Fig. 6** MgPar regions of P110. **a** Sequence alignment of the KL variable region of P110 and different MgPars encompassing the sialic binding site. KL region of MgPar-3 carries several STOP codons within the aligned segment and it was not included in the analysis. Similarly, the MgPar-6 region was omitted because it does not contain variable sequences homologous to P110. Residues shown to be critical for cell adhesion in this work (black star) appear to be fully conserved in the KL variable region of all the MgPar repeats. These residues are also conserved in the *M. pneumonia* protein MPN_142, orthologous to P110 (lower part of the panel). **b** Two views, 90° apart from each other, of the P110 surface. The KL (residues 42–294 in red) and LM (residues 462–517 in blue) variable regions and the poly-Serine tract (409–419 in yellow) of the P110 protein are depicted. Variable regions surround the receptor binding site where a bound oligosaccharide is shown (green)

and Pointless[57] from CCP4i suite of programs[58]. SeMet crystal belongs to the I222 space group with unit cell parameters **a** = 109.93 Å, **b** = 152.91 Å, **c** = 174.69 Å and one monomer in the asymmetric unit. The positions of all six expected Se atoms were located with ShelxC/D and phased up to 2.95 Å with ShelxE[59]. Initial phases were improved with 10 cycles of density modification with RESOLVE[60] and phases extended with data from a native crystal up to 2.73 Å. 3SL and 6SL data were solved by Molecular Replacement with native coordinates as an initial model to determine orientations and positions of one monomer in both 3SL and 6SL in the asymmetric unit by Phaser[61]. Final native, 3SL and 6SL models were manually traced and refined with cycles of Coot[62] and Refmac5[63], respectively.

**Surface plasmon resonance**. Binding kinetics were determined by surface plasmon resonance using a Biacore 3000 biosensor platform (GE Biosystems) equipped with a research-grade streptavidin-coated biosensor chip S1. The chip was docked into the instrument and preconditioned with three 1-min injections of 1 M NaCl in 50 mM NaOH. Both 3SL-PAA-biotin and 6SL-PAA-biotin (Carbosynth) oligosaccharides were injected over the second and third flow cell, respectively at 10 µg/ml diluted in HBS-P (10 mM Hepes, pH 7.4, 0.15 M NaCl and 0.005% P20). The first cell was left blank to serve as a reference. The running buffer consisted of HBS-P at a flow rate of 30 µl/min and the immobilisation levels acquired were ~150 and ~175 response units for 3SL-PAA-biotin and 6SL-PAA-biotin, respectively.

A series of diluted purified erP110 protein in HBS-P (0.31, 0.62, 1.25, 2.5, 5, and 10 µM) were injected over the flow cell surface at 30 µl/min. Interaction analysis

were performed at 25 °C and the protein was allowed to associate and dissociate for 120 and 240 s, respectively followed by a 30 s regeneration injection step of 0.05% SDS at 30 µl/min. Duplicates injections of each analyte and blank (buffer HBS-P) were used to produce data suitable for kinetic calculations. The data was fitted assuming a Langmuir 1:1 binding model using the steady state analysis option available within BiaEvaluation 3.1 software in order to determine the equilibrium dissociation constant $K_D$.

**Strains, culture conditions, and primers**. *M. genitalium* was grown in SP-4 medium at 37 °C in tissue culture flasks. Mutants were isolated on SP-4 agar plates supplemented with puromycin (3 µg mL$^{-1}$). All *M. genitalium* strains used in this work are listed in Supplementary Table 3. *E. coli* XL1-Blue strain was used for cloning and plasmid amplification purposes. It was grown in LB broth or agar plates containing 100 µg mL$^{-1}$ ampicillin. All primers used in this work are listed in Supplementary Table 2.

**DNA manipulation and mutant construction**. Plasmid DNA was purified using GeneJET Plasmid Miniprep Kit (Thermo Fisher Scientific). PCR products and DNA fragments were extracted from agarose gels using NucleoSpin Gel and PCR Clean-up Kit (Macherey-Nagel), and digested using the corresponding restriction enzymes (Fermentas) when necessary. For transformation of *M. genitalium*, plasmids were purified using the GenElute HP Midiprep Kit (Sigma) following the manufacturer's instructions.

First, we created a minitransposon (TnPac) carrying the puromycin selectable marker using the pMTn*TetM438* plasmid as scaffold[64]. The gene coding for the puromycin acetyl transferase (*pac*) was amplified from the pΔMG_218-lacZ plasmid[65] using *pac-F* and *pac-R* primers, and the resulting amplicon digested with *Eco*RI and *Bam*HI. The *tetM438* marker was excised from the pMTn*TetM438* plasmid with *Eco*RI and *Bam*HI, and similarly digested *pac* amplicon was ligated into the empty pMTn vector to create the pMTnPac plasmid. Then, we amplified the MG_192 gene from the chromosome of the *M. genitalium* G37 strain using COMmg192-F and COMmg192-R primers. The amplicon was digested with the *Apa*I and *Xho*I restriction enzymes and ligated to a similarly digested pMTnPac plasmid to create the pTnPacP110WT plasmid. This plasmid was created to reintroduce a wild-type allele of the MG_192 gene to the ΔMG_192 mutant. The COMmg192-F primer includes the upstream region (70 nucleotides) of the MG_191 gene, which contains a strong promoter identified in a previous study[66]. This promoter will drive transcription of the transposon-encoded copy of the MG_192 gene in all mutants.

Target mutations in the MG_192 gene were introduced using mutated oligonucleotides in a splicing by overlap extension PCR (SOE-PCR). To generate a S783A mutation, we PCR-amplified two overlapping segments encompassing the entire MG_192 gene using the COMmg192-F/S783Amg192-R and S783Amg192-F/COMmg192-R primers and DNA from the pMTnPacP110WT plasmid as a template. Then, both DNA fragments were joined in a SOE-PCR reaction using the COMmg192-F and COMmg192-R primers. The resulting fragment was cloned into a pMTnPac plasmid as described earlier to generate the pMTnPacP110S783A plasmid. A similar strategy was followed to generate pMTnPacP110W184A, pMTnPacP110S458D, and pMTnPacP110PP, and pMTnPacP110Δloop plasmids, using primers W184Amg192-F/R, PPmg192-F/R, loopmg192-F/R, and S458Dmg192-F/R, respectively. Sequencing analysis of the different TnPacP110 minitransposons using primers Tnp3, RTPCR192-F, RTPCR192-R, and PacUp, ruled out the presence of additional mutations within the MG_192 sequence. In addition, sequencing reactions confirmed the presence in the wild type as well as the mutant MG_192 alleles of the same number of AGT triplets, which code for a tract of ten serine residues found in the center of the P110 protein. Identification of the minitransposon insertion site in the mutants was done by sequencing using the PacDw primer and chromosomal DNA as a template.

**Transformation and screening**. *M. genitalium* ΔMG_192 mutant was transformed by electroporation using 5 μg of plasmid DNA of the different minitransposons[64]. Puromycin resistant colonies were picked, propagated and stored at −80 °C. First, we determined the insertion site of the minitransposon in these transformants. Insertions laid within the tetracycline resistance marker (*tetM*) in approximately half of the clones analyzed (Supplementary Figure 5). The *tetM* marker was previously used to create and select for the ΔMG_192 mutant. We hypothesize that clones carrying the MG_192 copies within the *tetM* marker have a superior fitness. However, we also identified minitransposon insertions in other genes such as MG_018, MG_238, MG_244, and MG_293 in other transformants. Of note, the location of the MG_192 mutant alleles in the same gene and genome context facilitates the phenotypical characterization and later comparison of the isolated mutants. For this reason, mutants carrying the TnPacP110 minitransposon within the *tetM* marker were selected for further analysis. For screening purposes, strains were further propagated in 25 cm² tissue culture flasks with puromycin and lysed using 0.1 M Tris-HCl pH 8.5, 0.05% Tween-20 and 250 μg mL$^{-1}$ Proteinase K for 1 h at 37 °C. Then, Proteinase K was inactivated at 95 °C for 10 min. *M. genitalium* lysates were screened for by sequencing using the PacUp and PacDown primers. On the other hand, the MG_192 alleles were fully re-sequenced to rule out the presence of undesired mutations.

**Sequencing reactions**. Sequencing reactions were performed with the BigDye® v3.1 Cycle Sequencing kit using 2.5 μL of genomic DNA, following the manufacturer's instructions. All sequencing reactions were analyzed using an ABI PRISM 3130xl Genetic Analyser at the Servei de Genòmica i Bioinformàtica (UAB).

**SDS–PAGE**. Whole cell lysates were obtained from mid-log phase cultures grown in 75 cm² flasks. Protein concentration was determined with the Pierce$^{TM}$ BCA Protein Assay Kit (Thermo Fisher Scientific) and similar amounts of total protein were separated by SDS–PAGE electrophoresis following standard procedures.

**Quantitative HA assay**. HA was quantified using flow cytometry as previously described[38] with few modifications. Binding of mycoplasma cells to red blood cells can be modeled in an inverse Langmuir isothermal kinetic function $\left(M_f = 1 - \frac{B_{max}[RBC]}{K_d + [RBC]}\right)$. $10^9$ mycoplasma cells were used during the HA assay. FACS data was acquired using a FACSCalibur (Becton Dickinson) equipped with an air-cooled 488 nm argon laser and a 633 nm red diode laser and analyzed with the CellQuest-Pro and FACSDiva software (Becton Dickinson). Where indicated, hRBCs ($2 \times 10^7$) were treated with 5 mU or 20 mU of neuraminidase from *Clostridium perfringens* (Merck) for 30 min at 37 °C and washed twice with PBS to remove the enzyme before the incubation with mycoplasma cells.

**Data availability**

Atomic coordinates and structure factors have been deposited in the PDB (www.rcsb.org) and the accession code assigned are 5MZ9 [https://www.rcsb.org/structure/5MZ9], 5MZD [https://www.rcsb.org/structure/5MZD] 5MZB [https://www.rcsb.org/structure/5MZB]. The data that support the findings of this study are available from the corresponding authors upon reasonable request.

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

## Acknowledgements

This work has been supported by grants BIO2013-48704-R, BFU2013-50176-EXP, BFU2015-71092-P, and BIO2017-84166-R from MINECO and MICINN (Spain). D.A. acknowledges María de Maeztu Unit of Excellence grant MDM-2014-0435 and S.T. acknowledges a PIF fellowship from the Universitat Autònoma de Barcelona. Many thanks are given to the XALOC beamline team at ALBA for all the support during data collection and, in particular, to Dr. Roeland Boer for his help during the resolution of the SeMet derivatives. We are also very grateful to Manuela Costa (UAB) for her valuable advice on performing FACS analysis and to Dr. Marta Taulés for her expert tips during SPR experiments.

## Author contributions

Conceived and designed the experiments: D.A., O.Q.P., and I.F. Performed the experiments: D.A., S.T., O.Q.P., and M.R. Analyzed the data: D.A., S.T., O.Q.P., J.P., and I.F. Contributed reagents/materials/analysis tools: E.Q., J.P. and I.F. Wrote the paper: D.A., O.Q.P., and I.F.

## Additional information

**Competing interests:** The authors declare no competing interests.

