## [Peer Review File · Nature Communications]

Reviewers' Comments:

Reviewer #1:

Remarks to the Author:

The manuscript describes an investigation into the molecular basis for the recognition of sialic acid receptors on host cells by the M genitalium P110 surface protein. The novelty lies principally in the structural characterisation of P110, which is a major adhesin of M genitalium. In several places the manuscript could have been better written and presented- I have noted some of these instances below but it requires further careful proof reading.

Major

Line 133 P110 and P140 together form a transmembrane complex (NAP) and it is this which was visualized by EM. How is it that the overall shape of erP110 is in good agreement with this lower resolution volume, when the P140 component is missing? How does this relate to the docking of P110 shown in Fig 6B- should this model not also include P140? How was the docking carried out?

Line 152 Is erP110 a dimer in solution? The text states that the C-domain forms extensive interactions across a 2-fold symmetry axis; has this interface been analyzed in PISA (<http://www.ebi.ac.uk/pdbe/pisa/>), for example? It would be useful to have some discussion on this point, if dimerization has any physiological significance.

Line 163 Is there any independent evidence that either of these two ligands bind to erP110? It would be good practice to verify this with a different method: ITC would provide a binding affinity and stoichiometry, and thus separate validation that these ligands bind.

Line 166 and Fig 2 Were any steps taken to eliminate phase bias from the (Fo-Fc) difference maps computed for this figure? Were these the strongest and most extensive segments of density observed in the difference maps? The quality of the density maps, particularly for 3SL, does not look that good, but possibly that is because they are generated before refinement. Improved justification for the fitting of these ligands to this difference density is warranted (and probably some clearer figures as well).

Line 177 Occupancy refinement at this resolution is not easy to justify. It could be the case that the B factors for the two ligands are different. The authors should consider an independent method for measurement of binding (see previous comments)

Line 203 Binding of K⁺ is commonly observed in many crystal structures. There seem to be minor structural changes to accommodate K⁺ binding but it is unclear whether these have any implications for the binding of oligosaccharide ligands. If K⁺ binding is important, it would be reflected in the binding affinities of the two oligosaccharides (through comparison of binding in different salts). Unless such evidence can be obtained, this section should be considerably shortened or omitted.

Line 207 Is the X-ray fluorescence data shown?

Line 272 What is the evidence that the binding assay shown in Fig 5C is measuring the interaction detected in the crystal structures of the erP110-oligosaccharide complexes? Without this, the effect of the W148A mutant could be attributable to indirect effects. It should be possible to use the 3SL or 6SL ligands to compete for binding in the assay, and thus act as inhibitors.

Minor

Line 21 spell out NAP abbreviation

Line 40 'sialylated'

Line 69 Why is this remarkable?

Line 73 rephrase this sentence- M genitalium does not comprise nine DNA repeat regions

Lines 94-95 Rather clumsy phrasing here

Line 178 '...in spite of the fact that...'

Line 185 'No structural similarities were found....'

Line 410 I think the authors mean one monomer in the asymmetric unit

Lines 412-414 This is written in a rather confusing manner- DM is density modification; why would it be implemented after RESOLVE which essentially does the same thing?

Reviewer #2:

Remarks to the Author:

General comments: In this study, Aparicio and colleagues crystallized and solved the structure of P110, one of the immunodominant adhesins of *Mycoplasma genitalium*, a sexually transmitted pathogen. They show that this protein is composed of a structural core of B-sheets, a hydrophobic "crown" that binds host cell sialic acid moieties, and a tail (with a hydrophobic region) that allows this protein to be inserted in the bacterial membrane. Using biochemical, genetic, and biologic techniques, they identify the specific amino acids of this protein that interact with sialic acid residues, characterized a sialic acid binding pocket, and demonstrate that P110 derivatives containing mutants of these amino acids are defective in binding red blood cells. This group also found that P110 binds potassium, changing the configuration of this protein. A beautiful model of the P100 protein is shown in Fig 6. In previous studies, many of which were published by this group, P110 has been shown to form a complex with P140 on the surface of Mgen's terminal organelle; together these proteins mediate adherence, motility, and replication. The work in this manuscript may be a first step in elucidating how these activities are accomplished. It will be interesting to see in future studies how this model relates to P140 to mediate these activities.

I found this paper extremely hard to read, in part due to (1) the extensive details presented in the text of the results, which "hide the forest for the trees," (2) the lack of clear labeling of the figures (esp Sup Fig 2), and (3) the need for summaries and cross references between the different figures and text descriptions. Better linkages between the results described in the text, displayed in the figures and the supplemental figures would help the readers as would summaries of the major findings in each section. Expanding the figure legends in the figure legends of the supplementary figures would be helpful.

Detailed comments, suggesting both edits and corrections for better English usage

L 49. Change "or" to "and;" M gen causes both cervicitis and pelvic inflammatory disease

L51-53. The possible intracellular localization of Mgen does not account for the frequent failure of our current antibiotic treatments, azithromycin, doxycycline, and moxifloxacin, which are active inside host cells and therefore would kill intracellular Mgen. I see no reference to this phenomenon in the references cited for this statement [Jensen (L53) and Rottem and Naot, 1998 (L291)] so these references should not be cited for this statement. In addition, by my reading of the current literature, the majority of MG cells are attached, but not inside host cells in contrast to the statement in this paragraph that implies that the adhesion is often (always) followed by invasion.

L 53. The word should be "resistance" rather than "reluctance."

L53. Suggest: Change "urge" for "emphasizes the urgency for"

L64. Delete "and"

L66. Suggest: Change "from" to "belong to"

L71 This would read better if it read: ...Ma et al, 2015... *M. genitalium* has evolved...

immunodominant proteins using recombination between these cytoadhesin genes and nine separate DNA repeat regions, designated as MgPars (Pearson et al, 1995). Note that the Pearson paper doesn't show that recombination is involved. That was shown by Burgos et al, 2012.

L80. The way the structural information guided the design of the P110 variants containing important cytoadherence defects is not very well outlined in the paper. There is so much detail that that the important epitopes and structure of the protein is hard to follow. I suggest rewriting the results part of the manuscript so the reader could more easily follow this great work.

L 83. This sentence would read better as: ...important clues to the understanding of the antigenic properties...

L89. Suggest that you change "of" to "from" i.e. Crystals were obtained from the...

L89. This sentence isn't clear. Also why is P110 considered the major adhesion? P140 is often designated as the primary adhesion and thus should be mentioned.

L-89-92. Doesn't make sense and needs to be rewritten

L133. Deleted "outer" Mycoplasmas don't have outer membranes. You probably mean that the C-terminal region of erP110 is close to the outer surface of its plasma membrane. Also not mentioned here, it might be important to add (speculate?) that the extreme C terminus of the native P110 protein is probably intracellular.

Line 139 The authors could point out that NAP structure presented in Scheffer et al is composed of both P110 and P140.

Line 11. The change in nomenclature for the P110 protein to MG192 is confusing. Would it be possible to always use one designation for P110/MG192 in the paper? Or at least when first using P110, remind the reader that this is encoded by the MG192 gene.

P148. Think "note" is a better word than "remark"

L280-281. This might be a good place to note that in addition to 184, 197/199 (I assume this is the PP) mutant playing a key role in attachment of MG with host cell receptors, structural analyses also implicates these proteins in sialic acid binding. Trying to summarize your data to help the readers.

L309. I'm unclear about the significance of proteins binding potassium. Is this common? Are there other potassium-binding proteins? If so, key references should be cited in this paper. It doesn't seem unusual to me that potassium binding would result in conformational changes in the protein similar to the activity of other ligands. I think you need a stronger argument to state that potassium binding results resulting hinge movements that mediating motility.

L326. A reference is needed for this statement about the localization of the sialic acid binding site in neuraminidases.

L315 This would read better as ...presents a topology consistent with a seven blade B-propeller that...

P275. Delete "mutant". P110-Wt. It is not a mutant

L370 This sentence should mention that Mgen is considered a "superbug" only because of its antibiotic resistance

Supplemental Fig 1. It may be interesting to note that the algorithm used to predict signal sequences of *E. coli*, this might not translate to the terminal organelle proteins of Mgen and *M. pneumo*. The N terminal portions of P140 and P1 are also annotated to have these short signal peptides, yet they appear to be much longer signal peptides (~50 AA from the translational start), see B Mader et al 1991 ("The mature MgPar adhesion of *M. genitalium*"), Iverson-Cabral, 2015 and M. Miyata and T. Hamaguchi, 2016). Although to my knowledge this hasn't been shown in P110 or P90/P40, I suspect, due to their co-localization and interaction with the P140 and P1 adhesins, these proteins might have a similar signal sequences. This might be worth mentioning. This figure also might be a good place to document the specific AA's included in erP110.

Supplemental Fig 1. I'm wondering if the predicted signal peptide of the P110 is correct. In the co-localized protein, P140, the N terminal of the mature protein is at AA 51 (B Mader et al, 1991); the signal peptide algorithms developed for walled proteins a different N terminus.

Supplemental Fig 2. Prior to this figure, P110 has been the designation for this protein, so if MG 192 is used here, its relationship to P110 should be noted (or at least included in parentheses).

This figure would be a good place to note the major landmarks cited in the text (and better define the ones that are listed). The landmarks indicated in the figure legend (black triangles, red stars, and black circles) were insignificant and hard to find, the symbols for the B strands and alpha helices (labeled B-1 to 53 and a1-a14), the designation of TT should be defined as should the red shading and the red lettering. The location of the B sheets II – VI should also be indicated.

Fig 1. The fact that P110 and the B barrel propeller from Virginiamycin B Lyase protein can be superimposed is not obvious to me from Fig 1. Could this be more clearly demonstrated?

Fig 4 (and Fig 5A). Could the authors describe what the different colors (red, blue, brown) represent? Also could the exact location of the different amino acids be depicted more clearly in this figure? ? Also for Fig B and FigC, the references for the binding of SprA and GspB should be cited, possibly by stating that these figures were "adapted from" Ref XX and XX.

Fig 3. The RBC binding activity of the P110 construct containing S783A mutation should be shown in Fig 5C. Does this mutation not overlap with the sialic acid binding sites? If so, it may be an important control for the other mutated proteins. Also it would be helpful to the readers if the locations of the residues in the binding pocket and the SNPs in the transposon P110 mutants S783A, W184A, PP, and S783A could be shown in Supplemental Fig 2 so the mutations in these constructs and the location of the sialic acid binding pockets could be visualized and perhaps linked.

Reviewer #3:

Remarks to the Author:

This paper studied about *Mycoplasma genitalium*, a human pathogen. The authors focused on P110, a major component of the adhesin complex, which is involved in adhesion, gliding motility, cell division, and antigenic variation. They clarified its 3D structure, suggested the mechanism of sialic acid binding, and proved it by genetic analyses.

The results are clear, useful and sometime surprising. The writing is also easy to understand. I have only some suggestions about presentation.

Supplemental Figure S2

This will be more useful if it includes additional information. Mark Blades I to VII, cleavage points in *M. pneumoniae*, N and C domains.

Figures 3 and 4

Renumber these.

L262 "the Δ loop variant could not be purified."

Did the authors try protein purification? Clarify this.

L271 "Binding of mycoplasma cells to red blood cells can be modeled in an inverse Langmuir isothermal kinetic function (Figure 5C)."

Show the function.

References

Scientific names should be italicized.

Figures 1, 2, 3

The crown and C-domain do not look like violet and brown. Please check the output color before publication.

L752 "Overall views of the potassium binding side..."

Is "side" correct?

L778 "(B) Docking of P110 into electron microscopy reconstructions from intact NAPs (15-20 Å resolution) (Scheffer et al., 2017),"

Is this really docking into the EM image? The image in the original paper looks more detailed.

Reviewer #1:

The manuscript describes an investigation into the molecular basis for the recognition of sialic acid receptors on host cells by the *M genitalium* P110 surface protein. The novelty lies principally in the structural characterization of P110, which is a major adhesin of *M genitalium*. In several places the manuscript could have been better written and presented- I have noted some of these instances below but it requires further careful proof reading.

The new text (including the modifications indicated below) has been revised by a native English speaking corrector.

Major

Line 133 P110 and P140 together form a transmembrane complex (NAP) and it is this which was visualized by EM. How is it that the overall shape of erP110 is in good agreement with this lower resolution volume, when the P140 component is missing? How does this relate to the docking of P110 shown in Fig 6B- should this model not also include P140? How was the docking carried out?

The low resolution volume of NAPs, provided by the Electron Microscopy studies, was described (Scheffer et al., 2017) as corresponding to a bulky globular region connected by a thinner region, a stalk, to the *M. genitalium* cell membrane. This description matched qualitatively very well the overall structural organization determined now for P110, with the large beta propeller and the small C-terminal domains, where the C-domain, which is followed in sequence by the transmembrane helix, is expected to be close to the cell membrane. This qualitative matching appears to be confirmed by a more quantitative fitting that we performed with program CHIMERA. In this fitting (sketched in the old Figure 6B) about half of the volume remains empty, which we assumed should correspond to the P140 proteins (the text has been changed -second paragraph in page 6- in an attempt to be clearer). However, in spite of this good overall fitting, we agree with the reviewer that the published low resolution EM volume of NAPs does not warrant certainty in the details of the fitting and it is likely better to omit old Figure 6B. The new figure and the new figure caption have been changed accordingly.

Line 152 Is erP110 a dimer in solution? The text states that the C-domain forms extensive interactions across a 2-fold symmetry axis; has this interface been analyzed in PISA (<http://www.ebi.ac.uk/pdbe/pisa/>), for example? It would be useful to have some discussion on this point, if dimerization has any physiological significance.

By gel filtration erP110 behaves as a monomer in solution. In agreement with this, the observed interactions between C-domains across the crystal 2-fold axis do not support the formation of stable dimers of erP110 in solution. The text has been changed to emphasize the monomeric character of erP110, eliminating the sentence about the crystal interactions of the C-domain (**Line142**) and also the lower panel in the old Figure 3.

Line 163 Is there any independent evidence that either of these two ligands bind to erP110? It would be good practice to verify this with a different method: ITC would provide a binding affinity and stoichiometry, and thus separate validation that these ligands bind.

We fully agree with the reviewer that an independent validation of binding in vitro was desirable. However, our attempts to perform ITC binding measurements with different erP110 constructs (and also with different ITC equipments) presented always experimental problems, apparently related with aggregation, that we were not able to overcome.

Now Surface Plasmon Resonance (SPR) experiments using biotinylated 3SL and 6SL immobilized on a streptavidin chip, provide a clear evidence that 3SL and 6SL bind to erP110 with μ molar affinities. Now, in the corrected manuscript, a new paragraph (L189-194) and panel A of the new Figure 4 have been added to present the outcome of these SPR experiments. Additionally, a SPR section has been included in Experimental Procedures (L384).

Line 166 and Fig 2 Were any steps taken to eliminate phase bias from the (Fo-Fc) difference maps computed for this figure? Were these the strongest and most extensive segments of density observed in the difference maps? The quality of the density maps, particularly for 3SL, does not look that good, but possibly that is because they are generated before refinement. Improved justification for the fitting of these ligands to this difference density is warranted (and probably some clearer figures as well).

Densities corresponding to oligosaccharides 3SL and 6SL were the largest blobs of unexplained density in the corresponding, sigma weighted, (2Fo-Fc) and (Fo-Fc) maps. The presence of these densities correlates with the co-crystallisation of 3SL or 6SL. Density was totally absent in the erP110 apo structures (both with and without potassium). Densities shown for 3SL and 6SL in Figure 2 correspond to standard omit maps (where oligosaccharides had been omitted from a model that was subjected to some refinement cycles) and consequently bias is expected to be (very) small. Density for 3SL certainly looks less well defined than for 6SL, likely due to either lower occupancy or higher mobility, or a combination of both, than 6SL (see below in the answer to the next query).

Line 177 Occupancy refinement at this resolution is not easy to justify. It could be the case that the B factors for the two ligands are different. The authors should consider an independent method for measurement of binding (see previous comments)

We agree with the reviewer that, as indicated before, quantification of the relative contribution of occupancy versus mobility it is not easy, in particular at the resolutions available, because both are highly correlated. Therefore, though it is clear in the electron density maps that density for 3SL looks weaker than for 6SL, we cannot decide unambiguously if a low occupancy is a

complete explanation. The SPR binding experiments of 3SL and 6SL to erP110 (mentioned before), an independent and quantitative estimations of affinity values, show that 3SL presents a lower affinity than 6SL, providing some support for the assignment of a low occupancy to 3SL. **(Fig.4A) (L189-194) (L384)**

Line 203 Binding of K⁺ is commonly observed in many crystal structures. There seem to be minor structural changes to accommodate K⁺ binding but it is unclear whether these have any implications for the binding of oligosaccharide ligands. If K⁺ binding is important, it would be reflected in the binding affinities of the two oligosaccharides (through comparison of binding in different salts). Unless such evidence can be obtained, this section should be considerably shortened or omitted.

We agree with the reviewer that there are no evidences that the structural changes observed to accommodate the binding of K⁺ could affect to the binding affinities of oligosaccharide ligands. Following the reviewer suggestion this section has been shortened **(L248)** and the lower panel in old Figure 3 omitted. The potassium structure has also been omitted from Supplemental table1.

Line 207 Is the X-ray fluorescence data shown?

The X-ray fluorescence data (strongly supporting the presence of potassium) is now included as supplementary Figure S4.

Line 272 What is the evidence that the binding assay shown in Fig 5C is measuring the interaction detected in the crystal structures of the erP110-oligosaccharide complexes? Without this, the effect of the W148A mutant could be attributable to indirect effects. It should be possible to use the 3SL or 6SL ligands to compete for binding in the assay, and thus act as inhibitors.

Again we have to agree with the observation of the reviewer that it was not proved that the hemadsorption assay used (old Fig 5C) was measuring the ability of Mycoplasma genitalium (Mge) cells to bind oligosaccharide sialylated complexes. Following the reviewer suggestion we have assessed the inhibitory effect of 6SL on Mge binding to human red blood cells (hRBCs). We observed an increase of K_D in the presence of 6SL, in agreement with a competitive inhibitory effect of 6SL on Mge hemadsorption. However, this K_D increment was not fully reproducible and it was observed only in some experiments. Unidentified factor/s of the *in vivo* hemadsorption assay might interfere with the ability of 6SL to compete with the binding of Mge to hRBCs. In the absence of a convincing explanation for these poorly reproducible results, we do not feel comfortable including these data in the manuscript.

We have designed an alternative experimental to demonstrate that the hemadsorption assay is testing the ability of Mge cells to bind sialylated oligosaccharides on the erythrocytes

membranes. First, we treated hRBCs with neuraminidase, an enzyme that removes very efficiently sialic acid moieties from the surface of hRBCs. Then, neuraminidase treated hRBCs were submitted to the hemadsorption assay to quantify the adhesion of Mge cells. The results of this new experimental were very reproducible and showed unequivocally that in the absence of sialic acid moieties, Mge cells were unable to adhere to hRBCs (**Figure 4D**). Therefore, the hemadsorption assay employed is measuring the ability of Mge cells to interact with sialic acids. The rationale of this new experimental and the results obtained are described in (**L228-233**) and (**L491-494**).

Minor

Line 21 spell out NAP abbreviation:

NAP is the actual name of the P110-P140 complex and it does not stand for any abbreviation. It is presented in capital letters for consistency with previous publications (Scheffert et al., Mol Micro., 2017).

Line 40 'sialylated': done (L 39)

Line 69 Why is this remarkable? Modified eliminating "remarkably" (L 64)

Line 73 Rephrase this sentence- M genitalium does not comprise nine DNA repeat regions

Changed as:

"Scattered throughout the M. genitalium genome there are nine repeat regions, designated as MgPars, that contain....." (**L65-67**)

Lines 94-95 Rather clumsy phrasing here

Changed as:

"The erP110 structure was solved by Single-wavelength Anomalous Diffraction (SAD) and density modification at 2.95 Å resolution (see experimental procedures) (Figure 1A). (**L86**)

Line 178 '...in spite of the fact that...' Done (L166)

Line 185 'No structural similarities were found....' Done (L173)

Line 410 I think the authors mean one monomer in the asymmetric unit. Changed. (L375)

Lines 412-414 This is written in a rather confusing manner- DM is density modification; why would it be implemented after RESOLVE which essentially does the same thing?
Programs RESOLVE and DM were both used in different attempts to improve the electron

density maps. However, as the reviewer indicates, the outcome was similar. The two sentences have been modified. (L376-377)

Reviewer #2:

General comments: *In this study, Aparicio and colleagues crystalized and solved the structure of P110, one of the immunodominant adhesins of Mycoplasma genitalium, a sexually transmitted pathogen. They show that this protein is composed of a structural core of B-sheets, a hydrophobic “crown” that binds host cell sialic acid moieties, and a tail (with a hydrophobic region) that allows this protein to be inserted in the bacterial membrane. Using biochemical, genetic, and biologic techniques, they identify the specific amino acids of this protein that interact with sialic acid residues, characterized a sialic acid binding pocket, and demonstrate that P110 derivatives containing mutants of these amino are defective in binding red blood cells. This group also found that P110 binds potassium, changing the configuration of this protein. A beautiful model of the P100 protein is shown in Fig 6. In previous studies, many of which were published by this group, P110 has been shown to form a complex with P140 on the surface of Mgen’s terminal organelle; together these proteins mediate adherence, motility, and replication. The work in this manuscript may be a first step in elucidating how these activities are accomplished. It will be interesting to see in future studies how this model relates to P140 to mediate these activities.*

I found this paper extremely hard to read, in part due to (1) the extensive details presented in the text of the results, which “hide the forest for the trees,” (2) the lack of clear labeling of the figures (esp Sup Fig 2), and (3) the need for summaries and cross references between the different figures and text descriptions. Better linkages between the results described in the text, displayed in the figures and the supplemental figures would help the readers as would summaries of the major findings in each section. Expanding the figure legends in the figure legends of the supplementary figures would be helpful.

The new text has been revised by a native English speaking corrector.

Cross references between text and Figures have been fully revised and the legends of supplementary figures have been expanded as suggested by the reviewer in an attempt to facilitate reading.

Detailed comments, suggesting both edits and corrections for better English usage
L 49. Change “or” to “and;” M gen causes both cervicitis and pelvic inflammatory disease

Done (L46)

L51-53. The possible intracellular localization of Mgen does not account for the frequent failure of our current antibiotic treatments, azithromycin, doxycycline, and moxifloxacin, which are active inside host cells and therefore would kill intracellular Mgen. I see no reference to this phenomenon in the references cited for this statement [Jensen (L53)

and Rottem and Naot, 1998 (L291)] so these references should not be cited for this statement. In addition, by my reading of the current literature, the majority of MG cells are attached, but not inside host cells in contrast to the statement in this paragraph that implies that the adhesion is often (always) followed by invasion.

This is an important observation; the text has been modified as follows:

“Upon interaction with host cell receptors, some *M. genitalium* cells invade the host cells by an unknown mechanism (Jensen et al 1994, Rottem and Naot 1998). Intracellular location of *M. genitalium* constitutes an effective strategy to avoid the host immune system and likely contributes to the establishment of persistent infections (McGowin CL et al., 2009).” (L46-49)

" which provides a safe harbor from host defenses (Mc Gowin CL et al 2009)" (L270-271)

L 53. The word should be “resistance” rather than “reluctance.”

The sentence has been modified as follows (L49-51):

“In addition, the rapid emergence of antibiotic resistance in *M. genitalium* (Bradshaw et al., 2006, Couldwell and Lewis, 2015, Jensen et al., 2008), emphasizes the urgency for development of new therapeutic strategies.

L53. Suggest: Change “urge” for “emphasizes the urgency for” Done (L50-51)

L64. Delete “and” Done (L58)

L66. Suggest: Change “from” to “belong to” Done (L60-61)

L71 This would read better if it read: ...Ma et al, 2015... *M. genitalium* has evolved... Done (L64)

immunodominant proteins using recombination between these cytoadhesin genes and nine separate DNA repeat regions, designated as MgPars (Pearson et al, 1995). Note that the Pearson paper doesn't show that recombination is involved. That was shown by Burgos et al, 2012.

The reference to Peterson et al. 1995 (see text below) gives credit to the identification of the DNA repeat regions within the genome of *M. genitalium*. As for the demonstration that recombination is involved in the DNA exchange between the MgPar regions and the cytoadhesin genes, we are confident that the two references provided in the manuscript are appropriate (Iverson-Cabral et al., 2007, Ma et al., 2007). The manuscript by Burgos et al. 2012 demonstrates the role of RecA in this process, but the implication of recombination was already demonstrated.

L65-68 “Scattered throughout the genome, *M. genitalium* comprises nine DNA repeat regions, designated as MgPar, that contain sequences with homology to the cytoadhesin genes (Peterson et al., 1995). Recombination between the cytoadhesin genes and homologous MgPar sequences, provides a virtually unlimited collection of antigenic variants (Iverson-Cabral et al., 2007, Ma et al., 2007).”

L80. *The way the structural information guided the design of the P110 variants containing important cytoadherence defects is not very well outlined in the paper. There is so much detail that that the important epitopes and structure of the protein is hard to follow. I suggest rewriting the results part of the manuscript so the reader could more easily follow this great work.*

The text has been extensively revised; we really hope it is now clearer.

L 83. *This sentence would read better as: ...important clues to the understanding of the antigenic properties...*

Done (L76-77)

L89. *Suggest that you change “of” to “from” i.e. Crystals were obtained from the...* Done (L82)

L89. *This sentence isn’t clear. Also why is P110 considered the major adhesion? P140 is often designated as the primary adhesion and thus should be mentioned.*

Changed (L82)

L-89-92. *Doesn’t make sense and needs to be rewritten*

The sentence has been modified as two new sentences:

Crystals were obtained from the extracellular region of *M. genitalium* adhesion P119 (erP110) spanning from residue 23 to 938. Therefore, erP110 does not include either residues corresponding to the signal secretion peptide (residues 1-22), or the transmembrane and intracellular regions (938-1052) at the N- and C-ends of P110, respectively (Supplemental Figure S1)." (L82-86)

L133. *Deleted “outer” Mycoplasmas don’t have outer membranes. You probably mean that the C-terminal region of erP110 is close to the outer surface of its plasma membrane. Also not mentioned here, it might be important to add (speculate?) that the extreme C terminus of the native P110 protein is probably intracellular.*

The sentence has been modified as: “...is expected to be close to the outer surface of the mycoplasma membrane (Figure 2A)." (L123-124)

That the extreme C terminus is probably intracellular is already indicated in the answer to the previous query and in the Supplemental Figure S1

Line 139 *The authors could point out that NAP structure presented in Scheffer et al is composed of both P110 and P140.*

The sentence has been changed to include the reviewer suggestion as:

“This overall shape of NAPs, composed of both P110 and P140, can correspond with the organization of erP110 having the large, β -propeller, domain and the small and elongated C-domain.” (L127-130)

Line 11. *The change in nomenclature for the P110 protein to MG192 is confusing. Would it be possible to always use one designation for P110/MG192 in the paper? Or at least when first using P110, remind the reader that this is encoded by the MG192 gene.*

The designations P110 and MG_192 refer to the standard nomenclature for the adhesin protein and gene (locus tag), respectively. In the current version, we have been very careful with this distinction throughout the text and Figures. For better clarity, we have now included a reminder that the MG_192 gene codes for the P110 protein (L199). The text reads as follows:

“To assess the implication of these residues in cytoadherence, we performed target mutagenesis of the MG_192 gene, which codes for the P110 protein, to generate the following variants: W184A, S458D, S783A, P197A-P199A (double PP mutant) and deletion of the loop S458-T462 (Δ loop mutant).”

L148. *Think “note” is a better word than “remark”* Changed (L138)

L280-281. *This might be a good place to note that in addition to 184, 197/199 (I assume this is the PP) mutant playing a key role in attachment of MG with host cell receptors, structural analyses also implicates these proteins in sialic acid binding. Trying to summarize your data to help the readers.*

As suggested by the reviewer, we have modified the text as follows (L239-241): “Therefore, as indicated by the structural data, our *in vivo* analysis corroborates that Trp184, Pro197 and Pro199 play a key role in the interaction of *M. genitalium* with host cell receptors.”

L309. *I’m unclear about the significance of proteins binding potassium. Is this common? Are there other potassium-binding proteins? If so, key references should be cited in this paper. It doesn’t seem unusual to me that potassium binding would result in conformational changes in the protein similar to the activity of other ligands. I think you need a stronger argument to state that potassium binding results resulting hinge movements that mediating motility.*

Binding of potassium has a diversity of proved roles in different proteins. Two general references have been now added. (L260-261) We agree with the reviewer that the possible link between potassium binding and hinge movements would require a clearer experimental

support. The text has been shortened indicating the uncertainty about possible roles of potassium in P110.

L326. A reference is needed for this statement about the localization of the sialic acid binding site in neuraminidases.

The reference of Gaskell et al., 1995 has been added. (L299)

L315 This would read better as ...presents a topology consistent with a seven blade B-propeller that...

Done (L289-290)

L275. Delete “mutant”. P110-Wt. It is not a mutant

We have replaced “mutant” by “strain” (L233-235)

L370 This sentence should mention that Mgen is considered a “superbug” only because of its antibiotic resistance

Done (L335-336)

Supplemental Fig 1. It may be interesting to note that the algorithm used to predict signal sequences of *E. coli*, this might not translate to the terminal organelle proteins of *Mgen* and *M. pneumo*. The N terminal portions of P140 and P1 are also annotated to have these short signal peptides, yet they appear to be much longer signal peptides (~50 AA from the translational start), see B Mader et al 1991 (“The mature MgPar adhesion of *M. genitalium*), Iverson-Cabral, 2015 and M. Miyata and T. Hamaguchi, 2016). Although to my knowledge this hasn’t been shown in P110 or P90/P40, I suspect, due to their co-localization and interaction with the P140 and P1 adhesins, these proteins might have a similar signal sequences. This might be worth mentioning. This figure also might be a good place to document the specific AA’s included in erP110.

Supplemental Fig 1. I’m wondering if the predicted signal peptide of the P110 is correct. In the co-localized protein, P140, the N terminal of the mature protein is at AA 51 (B Mader et al, 1991); the signal peptide algorithms developed for walled proteins a different N terminus.

We thank the reviewer for the comments about the signal peptide. Paper “Experimental Proof for a Signal Peptidase I like Activity in *Mycoplasma pneumoniae*, but Absence of a Gene Encoding a Conserved Bacterial Type I SPase” (Catreine et al 2005), describes several ways to demonstrate the cleavage site for the signal peptide in P40/P90. Computational methods (SignalIP 3.0 program) are in agreement with experimental data for the prediction of the cleavage site between residues 25 and 26.

We also used the SignalIP 4.1 (Nielsen et al 1997) program with a cutoff of 0.34 to find the position of the signal peptide cleavage in P110. In agreement with the Psi-pred server, SignalIP 4.1 showed that the most likely cleavage position is between residues 21 and 22.

Additionally, following the reviewer suggestion, we have added in Supplemental Fig1 the specific aminoacids included in erP110

Supplemental Fig 2. Prior to this figure, P110 has been the designation for this protein, so if MG 192 is used here, its relationship to P110 should be noted (or at least included in parentheses). This figure would be a good place to note the major landmarks cited in the text (and better define the ones that are listed). The landmarks indicated in the figure legend (black triangles, red stars, and black circles) were insignificant and hard to find, the symbols for the B strands and alpha helices (labeled B-1 to 53 and a1-a14), the designation of TT should be defined as should the red shading and the red lettering. The location of the B sheets II – VI should also be indicated.

The new supplemental Figure S2 incorporates all the suggestions from the reviewer.

Fig 1. The fact that P110 and the B barrel propeller from Virginiamycin B Lyase protein can be superimposed is not obvious to me from Fig 1. Could this be more clearly demonstrated?

A stereo view of the superimposition is now presented as supplemental Figure S3.

Fig 4 (and Fig 5A). Could the authors describe what the different colors (red, blue, brown) represent? Also could the exact location of the different amino acids be depicted more clearly in this figure? ? Also for Fig B and FigC, the references for the binding of SprA and GspB should be cited, possibly by stating that these figures were “adapted from” Ref XX and XX.

Old Figure 4 corresponds to the new Figure 3. Colors of atoms are now described in the figure caption. The references to SprA and GspB are also included in the figure caption as suggested by the reviewer. Old figure 5A has been eliminated, because it is partially redundant with the new Figure 3A.

Fig 3. The RBC binding activity of the P110 construct containing S783A mutation should be shown in Fig 5C. Does this mutation not overlap with the sialic acid binding sites? If so, it may be an important control for the other mutated proteins. Also it would be helpful to the readers if the locations of the residues in the binding pocket and the SNPs in the transposon P110 mutants S783A, W184A, PP, and S783A could be shown in Supplemental Fig 2 so the mutations in these constructs and the location of the sialic acid binding pockets could be visualized and perhaps linked.

Indeed, the mutation S783A does not overlap with the sialic acid binding sites. In the text, we stated that “the hemadsorption capacity of the P110-S783A mutant was comparable to that of the P110-WT strain (data not shown).” As it can be seen in the image below (provided for the reviewers), the hemadsorption capacity of the S783A mutant overlaps with that of the P110-WT and G37 strains and it actually obfuscates the data from these two key strains. For this reason, we still think that it is preferable to omit the S783A data from the Figure (now Figure 4C). In the new supplemental Figure S2 residues in the sialic binding site are explicitly indicated. Mutated residues can be located with respect to these residues.

Reviewer #3:

This paper studied about Mycoplasma genitalium, a human pathogen. The authors focused on P110, a major component of the adhesin complex, which is involved in adhesion, gliding motility, cell division, and antigenic variation. They clarified its 3D structure, suggested the mechanism of sialic acid binding, and proved it by genetic analyses.

The results are clear, useful and sometime surprising. The writing is also easy to understand. I have only some suggestions about presentation.

Supplemental Figure S2: This will be more useful if it includes additional information. Mark Blades I to VII, cleavage points in M. pneumoniae, N and C domains.

As suggested by the reviewer, blades are indicated in the new supplemental Figure S2. The sentence referring to the cleavage point in M. pneumoniae has been omitted (L318).

Figures 3 and 4. Renumber these.

Figures have been renumbered and the references crosschecked.

L262 “the Δ loop variant could not be purified.” Did the authors try protein purification? Clarify this.

Yes, we unsuccessfully tried to purify the Δ loop P110 variant. By gel filtration chromatography, this protein sample showed a heterogeneous distribution of aggregates. The data is now included in the text (L218-219).

L271 “Binding of mycoplasma cells to red blood cells can be modeled in an inverse Langmuir isothermal kinetic function (Figure 5C).” Show the function.

The function is now shown in the “Experimental Procedures” section (L488).

References

Scientific names should be italicized.

The format of the References required by Nature Communications has been revised.

Figures 1, 2, 3 The crown and C-domain do not look like violet and brown. Please check the output color before publication.

The names of colors have now been crosschecked and modified when discrepancies were found.

L752 “Overall views of the potassium binding side...” Is “side” correct? Changed (L749)

L778 “(B) Docking of P110 into electron microscopy reconstructions from intact NAPs (15-20 Å resolution) (Scheffer et al., 2017),”Is this really docking into the EM image? The image in the original paper looks more detailed.

As explained also in the first query from reviewer #1, the fitting was performed with program CHIMERA and this fitting, where about half of the volume remains empty probably because P140 is missing, was only sketched in the old Figure 6B. However, as suggested by reviewer #1, the low resolution of the EM volume of NAPs published does not warrant certainty in the details of the fitting and it is likely better to omit the old Figure 6B. The new figure and the new figure caption have been changed accordingly.

Reviewers' Comments:

Reviewer #1:

Remarks to the Author:

My original comments have now been addressed, particularly around the issues of ligand binding. I have only one minor comment about the revision:

Lines 128-130 It would help to justify this statement with some estimates of overall dimensions of the volume from the EM reconstruction and comparison with the crystal structure- it is not just about shape.

Reviewer #2:

Remarks to the Author:

This is a very nice study characterizing the a unique protein in the attachment organelle of *Mycoplasma genitalium*. Insights into the structure of this molecule will undoubtedly provide insights into the enigmatic attachment and enigmatic motility of this organism and hopefully will inform future studies providing therapeutics for this important sexually transmitted pathogen.

Although all my concerns and edits have been addressed or incorporated into this current version, I have a few more comments that might improve the current version of this paper:

1. I agree with reviewer #1 that designation of NAP for the transmembrane complex of P140 and P110 should be defined. As written, it seems like the letters probably should stand for something, but they don't. The word defines "the raised surface of certain cloth, such as flannel" (see Wikipedia) or a "hairy or downy surface on a woven fabric " (Webster's dictionary) and was first used to aptly define the appearance these molecules, unknown at the time, decorating the outer surface of the attachment organelle seen by EMs in the first description of this organism, Tully et al (1981 and 1983). Noting this definition (and crediting Tully for this clever designation) should be included in the manuscript. I would help/entertain the non-English speaking (or not clothing-obsessed) people to see the origin of this designation
2. Line 96. Change "till" to "until."
3. Line 732. Do the authors mean "amino acid" residues instead of "protein?"
4. The figure on the right should be labelled "B"

Reviewer #1 (Remarks to the Author):

My original comments have now been addressed, particularly around the issues of ligand binding. I have only one minor comment about the revision:

Lines 128-130 It would help to justify this statement with some estimates of overall dimensions of the volume from the EM reconstruction and comparison with the crystal structure- it is not just about shape.

As suggested by the referee the paragraph has been modified to include a reference to both the shape and the size of the reconstructed Nap volumes. L-132-138

Reviewer #2 (Remarks to the Author):

This is a very nice study characterizing the a unique protein in the attachment organelle of Mycoplasma genitalium. Insights into the structure of this molecule will undoubtedly provide insights into the enigmatic attachment and enigmatic motility of this organism and hopefully will inform future studies providing therapeutics for this important sexually transmitted pathogen. Although all my concerns and edits have been addressed or incorporated into this current version, I have a few more comments that might improve the current version of this paper:

1. I agree with reviewer #1 that designation of NAP for the transmembrane complex of P140 and P110 should be defined. As written, it seems like the letters probably should stand for something, but they don't. The word defines "the raised surface of certain cloth, such as flannel" (see Wikipedia) or a "hairy or downy surface on a woven fabric " (Webster's dictionary) and was first used to aptly define the appearance these molecules, unknown at the time, decorating the outer surface of the attachment organelle seen by EMs in the first description of this organism, Tully et al (1981 and 1983). Noting this definition (and crediting Tully for this clever designation) should be included in the manuscript. I would help/entertain the non-English speaking (or not clothing-obsessed) people to see the origin of this designation

We agree with the reviewer that crediting Dr. Joseph Tully and colleagues for the original designation of the Nap is both worth and deserved.

The following sentence has been introduced in L60:

"a polar structure known as the terminal organelle. The term Nap was originally coined by Dr. Tully and colleagues to designate the short fibers decorating the outer surface of the terminal organelle of *M. genitalium* (Tully *et al.*, 1983)"

2. Line 96. Change “till” to “until.” L101 Changed

3. Line 732. Do the authors mean “amino acid” residues instead of “protein?” L753 Changed

4. The figure on the right should be labelled “B” The Figure 3 panel B is labelled with "B"